# Liquid Biopsy in Diagnosis and Prognosis of Non-Metastatic Prostate Cancer

**DOI:** 10.3390/biomedicines10123115

**Published:** 2022-12-02

**Authors:** Alexey S. Rzhevskiy, Alina Y. Kapitannikova, Denis V. Butnaru, Evgeniy V. Shpot, Simon A. Joosse, Andrei V. Zvyagin, Majid Ebrahimi Warkiani

**Affiliations:** 1Institute of Molecular Theranostics, Sechenov First Moscow State Medical University, 119991 Moscow, Russia; 2Faculty of Computer Science, National Research University Higher School of Economics, 101000 Moscow, Russia; 3Institute for Urology and Reproductive Health, Sechenov University, 119991 Moscow, Russia; 4Department of Tumor Biology, University Medical Center Hamburg-Eppendorf, Martinistr. 52, 20246 Hamburg, Germany; 5Mildred Scheel Cancer Career Center HaTriCS4, University Medical Center Hamburg-Eppendorf, Martinistr. 52, 20246 Hamburg, Germany; 6MQ Photonics Centre, Macquarie University, Sydney 2109, Australia; 7School of Biomedical Engineering, University of Technology Sydney, Sydney 2007, Australia

**Keywords:** prostate cancer, liquid biopsy, circulating tumor cells, ctDNA, cfDNA, exosomes, miRNA

## Abstract

Currently, sensitive and specific methods for the detection and prognosis of early stage PCa are lacking. To establish the diagnosis and further identify an appropriate treatment strategy, prostate specific antigen (PSA) blood test followed by tissue biopsy have to be performed. The combination of tests is justified by the lack of a highly sensitive, specific, and safe single test. Tissue biopsy is specific but invasive and may have severe side effects, and therefore is inappropriate for screening of the disease. At the same time, the PSA blood test, which is conventionally used for PCa screening, has low specificity and may be elevated in the case of noncancerous prostate tumors and inflammatory conditions, including benign prostatic hyperplasia and prostatitis. Thus, diverse techniques of liquid biopsy have been investigated to supplement or replace the existing tests of prostate cancer early diagnosis and prognostics. Here, we provide a review on the advances in diagnosis and prognostics of non-metastatic prostate cancer by means of various biomarkers extracted via liquid biopsy, including circulating tumor cells, exosomal miRNAs, and circulating DNAs.

## 1. Introduction

In clinical practice, prostate cancer (PCa) is currently diagnosed by performing prostate-specific antigen (PSA) blood tests and if the PSA serum levels are increased, followed by tissue biopsy and pathological inspection. However, the PSA serum levels are a relatively weak indicator of PCa due to its only moderate specificity [1]. Thus, a value of a PSA serum level at >4 ng/mL, recommended by most clinical guidelines as suspicious for PCa and therefore requiring the subsequent performance of a tissue biopsy, is often recognized in patients with non-cancerous lesions of the prostate including benign prostatic hyperplasia (BPH) [2] and prostatitis [3]. Furthermore, the poor effectiveness of the PSA serum level monitoring has been considered one of the most prominent causes of diagnosis of PCa at an advanced stage [4]. It is also worth noting that, in most countries, routine PSA screening is recommended for men in their 55–69 years only to avoid overdiagnosis in other age categories [5], which increases the risk of missing PCa in men under 55 years [6]. Thus, to increase the probability for diagnosing PCa at its early form, more specific and sensitive alternatives to the PSA blood test are required. The 12-core tissue biopsy [7] is the gold standard for PCa diagnosis; however,, even though the test has a specificity close to 99%, its sensitivity is relatively low [8] and requires ultrasound or magnetic resonance imaging guidance for better precision, which significantly increase financial cost of the procedure [9]. Approximately only 40–50% of patients with elevated PSA serum level are tested positive for PCa by tissue biopsy [10]. Moreover, the outcome of the procedure is highly dependent on the experience of the operator, which compromises its objective diagnostic value. From the patient’s perspective, tissue biopsy is a highly invasive, painful procedure, having possible side effects such as local hematomas, abscesses, and even erectile dysfunction [11].

The most frequent initial tactics for managing localized prostate cancer are active surveillance and watchful waiting [12]. These tactics imply monitoring of PCa after its diagnosis by performing periodic PSA tests and tissue biopsies. In the case of cancer progression, the strategy changes to prostatectomy. However, the tactics of active surveillance and watchful waiting are not always appropriate due to genetic heterogeneity of the disease, and variety of its forms from indolent localized to rapidly developing [13]. Thus, to increase specificity and sensitivity of PCa detection at an early stage, to provide more accurate risk stratification and prognosis of the disease, timely identify the rapidly progressing cases, and to therefore develop more efficient treatment approaches, liquid biopsy via isolation and analysis of circulating biomarkers has been proposed [14].

To date, liquid biopsy has been considered one of the most promising techniques that could potentially replace or substitute to the existing PCa diagnostic and prognostic tests. Liquid biopsy techniques may be stratified in terms of the biomarker of interest, which are: CTCs, extracellular vesicles (EVs) or exosomes, proteins, tumor-educated platelets (TEPs), and circulating nucleic acids [15]. Thus, liquid biopsy has been termed as sampling of biological fluids containing any of these biomarkers and has been investigated as a minimally or non-invasive technique for cancer diagnostics and prognostics. As a potential technique for diagnosis and prognosis of malignant tumors, liquid biopsy was first described in literature more than a hundred years ago [16]. However, the technique has not been broadly investigated until recently due to a lack of approaches that would allow efficient isolation, identification, and analysis of CTCs. In the past two decades, the development of nano- and micro-technologies led to significant evolution of liquid biopsy via isolation of CTCs from diverse biological fluids, and resulted in discovery of novel liquid biopsy methodologies based on isolation of EVs and nucleic acids [17] (Figure 1).

CTCs are the cells that are shed into diverse biological fluids of a human body from primary or secondary tumor sites, and therefore may be detected in cancer patients while undetectable in healthy individuals [18]. The CTCs are vastly heterogeneous, first of all in terms of antigen profile and gene mutations [19]. Cell-free ctDNA are the fragments of DNA, which release into the body fluids due to lysis or apoptosis of tumor cells, or exocytosis [20,21]. Exosomes are the extracellular vesicles which are formed as a result of cell membrane blebbing and are released from tumor cells during their functioning. These vesicles are bearing various agents, including miRNAs, which may be recognized as tumor biomarkers [22]. The listed biomarkers are identified in such biological fluids of PCa patients as blood, urine, and semen [23]. The aim of this systematic review was to compile the existing data within the research field of liquid biopsy as the technique for early diagnosis and prognosis of PCa, to analyze the collected data, and to provide potential directions for investigations which would allow to implement the technique in clinic.

## 2. Search Strategy

The current review was prepared by selecting research articles, written in English, from two data bases: PubMed and Web of Science. To identify original articles which report on the application of diverse liquid biopsies for diagnosis and prognostics of localized PCa, article search was performed by using such key terms or their combinations as: “localized prostate cancer”, “non-metastatic prostate cancer”, “circulating tumor cells” or “CTCs” or “circulating cancer cells”, “exosomes” or “extracellular vesicles”, “circulating miRNA(s)”, “circulating DNA(s)”, “circulating RNA(s)”, “liquid biopsy”. Further, a filter for the publication dates from 1 January 2010 to 30 September 2022 was applied. Books, review articles, letters to editor, case studies, studies of PCa animal models or cell lines, and research articles that did not contain data on diagnostic and prognostic potential of liquid biopsy biomarkers in management of localized PCa were excluded from the analysis. Notably, from the research articles chosen for the current review, the data for both localized (T1 and T2 clinical stages) and locally advanced (T3, T4 clinical stages) forms of PCa was included.

## 3. Cell-Free DNA (cfDNA)

For the first time, cfDNA was isolated from human blood in 1948 [24]. DNA originating from the tumor can provide information about the genomic make-up, such as point-mutations and chromosomal aberrations, but also epigenetic changes that can be used as potential biomarkers for the diagnosis and monitoring of cancer development [25,26] (Figure 2). Among the entire amount of cfDNA, a small content called circulating tumor DNA (ctDNA) corresponds to a specific cancer-related DNA bearing the most significant diagnostic and prognostic information [27]. The average length of the ctDNA fragments is approximately 167 and 320 base pairs [21]. The shorter fragments coming from apoptotic degradation of the cellular DNA and the longer fragments as a result of necrosis in the tumor [28]. ctDNA is one of the components of extracellular DNA originating from tumor cells, and its total fraction is mainly dependent on the tumor stage [29]. Although rare, longer ctDNA fragments of >1 kb have recently been investigated as well for the identification of methylation profiles in hepatocellular carcinoma patients [30].

One of the main features of ctDNA is the ability to contain the same mutations that can be found in tumor tissue [31], including AR mutation, BRCA2 mutation, TMPRSS2:ERG fusion, PTEN loss or mutation, TP53 inactivation, and MYC amplification [32]. Another significant issue of ctDNAs, as diagnostic and prognostic biomarkers in PCa, is that the molecular landscape of the tumor, which is significantly variable depending on its stage [13]. It was demonstrated that there is a direct correlation between the stage of the disease, the amount of circulating cf(DNA) in biological fluids, and its variability. In the case of the localized form of PCa, the total amount of cell-free DNA in blood plasma may be relatively small [33] and is often insufficient to be determined even by modern, highly sensitive techniques. Previously, detection and analysis of mutations in small amounts of ctDNA, presenting as one of the fractions of extracellular DNA, posed a severe technical challenge [34]. However, in the last ten years, rising advancements in the techniques of ctDNA detection and analysis have led to broad investigation of ctDNA, predominantly from blood plasma, as potential diagnostic and prognostic biomarkers [21]. At the same time, although blood has been most widely investigated as the matrix for liquid biopsy, alternative biological fluids have recently gained significant attention [35,36]. Particularly in the case of urine, PCa biomarkers may be detected earlier, compared to blood [37]. In urine, ctDNA are exposed to less aggressive effects of damaging agents than those found in blood [38]. In addition, semen is the other perspective source of liquid biopsy biomarkers, which has not been studied thoroughly so far [39].

Gradual development of genetic heterogeneity is the distinctive feature of PCa evolution [40]. It is followed by the dominance of subclones, which promote metastasis and drug resistance. In the case of PCa, heterogeneity was discovered by pathologists in 1970s: it was found that diverse subtypes of PCa differ in nuclear morphology, cell proliferation, levels of immune cell infiltration, and several other features [41]. In addition to morphological heterogeneity, heterogeneous copy number variations were reported for the primary tumor [42]. Similarly, somatic single-nucleotide polymorphism was demonstrated among both localized and metastatic tumors [43]. Tracking heterogeneity in the mutational landscape of the tumor made it possible to identify the evolution of clones within individual PCa patients [44]. In its turn, ctDNA may be an object for identifying clinically relevant genomic alterations and tumor subclones. Thus, in the study by Chen et al. [45], somatic tissue alterations were identified in ctDNA derived from the blood plasma of PCa patients with localized prostate cancer, in 57% of cases, including such alterations as nonsynonymous variants in FOXA1, PTEN, MED12, and ATM. 

It is known that cancer is accompanied by initial hypomethylation of DNA and further hypermethylation in the promoter regions of tumor suppressor genes, which leads to inactivation of transcription and translation of the corresponding protein complexes [46]. DNA hypermethylation is one of the earliest and most common aberrations in PCa [47]. In the study by Brikun et al. [48], a panel of 32 epigenetic biomarkers of PCa was developed. Using methylation-sensitive qPCR, biopsy samples of the prostate glands of 104 patients were analyzed. The prostate tissues were investigated by using 24 well-known DNA methylation biomarkers. According to the results, significant difference was demonstrated between tumor and unaffected tissue in individual patients, and between PCa patients and the control group. In the further study by Brikun et al. [49], ctDNA was isolated from the urine of 94 patients. The expression levels of biomarkers, previously found in tissues, were analyzed, and compared with the data obtained from tissue biopsies. A specificity of the proposed technique was identified at 76%, with the sensitivity at 81%. In the other study by the same group of authors [50], a panel of DNA methylation markers suitable for a non-invasive diagnostic test for ctDNA in urine was identified. The urine was collected after a digital rectal examination (DRE) [51] and/or after a first morning void (FV). A specificity at 71% for both DRE and FV, and a sensitivity at 89% for DRE and at 94% for FV, was reported. In the study by O’Reilly et al. [52], the epiCaPture multi-biomarker panel was tested. Numerical alterations in DNA hypermethylation at the 5′ end of 6 genes (GSTP1, SFRP2, IGFBP3, IGFBP7, and PTGS2) were the targets for the panel. Previously, all these targets were identified as potential PCa biomarkers [53]. More than 450 urine samples of patients with elevated PSA were analyzed. According to the results, a higher degree of DNA methylation correlated with an increased risk of developing an aggressive form of PCa. It was also demonstrated that the specificity of the epiCaPture test was superior to the PSA screening. At the same time, as in was demonstrated by Bjerre et al., in the case of blood plasma, hypermethylation of ctDNA for DOCK2, HAPLN3, and FBXO30 genes, was not detected in neither of the participants: patients with localized PCa, BPH patients, or healthy volunteers [54].

Despite the existing positive data that determine ctDNA from blood plasma as a perspective biomarker for PCa early diagnosis and prognosis [55], different studies report on difficulties in ctDNA extraction from blood plasma of PCa patients predominantly due to its low concentration. Thus, in the study by Hennigan et al., ctDNA could not have been detected using ultra-low-pass whole-genome sequencing and targeted resequencing in blood plasma of patients with localized PCa, even in the case of the clinically high-risk patients. It has also been proven that concentration of ctDNA in blood plasma in patients with localized PCa may be significantly increased due to inflammation or traumatization of prostate, including traumatization due to tissue biopsy [56]. At the same time, even though it may not be possible to differentiate between PCa and healthy status based on cfDNA concentration in blood plasma, the cfDNA fragment size may be a reliable biomarker [57].

Recently, promising data has been obtained in the studies on isolation of cfDNA from the seminal fluid, which, in comparison to blood, contains much higher concentrations of cfDNA [58]. If comparing concentrations of cfDNA in seminal fluid of healthy volunteers or patients with BPH and patients with localized PCa, approximately 10-fold difference was identified [58,59]. However, despite great potential of seminal fluid as the source of PCa biomarker, limited number of investigations have been performed on identifying the efficacy of seminal cfDNA as a biomarker of localized PCa. 

In conclusion, the use of circulating extracellular and tumor DNA has great potential as a biomarker for early detection and stratification of PCa. Nevertheless, effectiveness and feasibility of using cfDNA as a biomarker of PCa has not yet been determined. Currently, various techniques have been used to investigate cfDNA, including ctDNA, with different, sometimes completely opposite, results.

## 4. Exosomal miRNAs

Exosomes are small EVs secreted by both normal and cancer cells. Exosomes are found in almost all types of cells and were successfully isolated from blood, urine, semen, and other body fluids [60]. Exosomes are released into the extracellular space via exocytosis and may carry different contents, such as proteins and different types of RNA and DNA, and function as intercellular messengers transmitting signaling molecules from one cell to another [61] (Figure 3). A large body of evidence suggests a critical role of the exosomes in tumor progression and metastasis. Another important feature of the exosomes is formation of drug resistance by inducing a tumor-friendly microenvironment due to involvement of the exosomes in proliferation, differentiation, and migration of fibroblasts, influence on immune cells, and stimulation of tumor angiogenesis. In its turn, exosomal miRNAs are of particular interest due to their ability to affect cancer progression by reducing apoptosis in a tumor and by increasing proliferation, migration, and adhesion of cancer cells [62,63,64]. At the same time, one of the main factors indicating the critical role of exosomal miRNAs in PCa carcinogenesis is their role in the regulation of epithelial–mesenchymal transition of PCa cells [65].

In recent years, exosomes containing miRNAs have been widely investigated as potential biomarkers of various malignant tumors, due to their functional relationship with tumor progression, including cell growth, differentiation, proliferation, angiogenesis, and apoptosis [66,67,68]. Extracellular miRNAs have been isolated from various biological fluids, such as blood, urine, saliva, breast milk, and semen [69]. They were found both inside extracellular vesicles, such as exosomes, and in free form [70,71]. It was identified that extracellular miRNAs influence targeted cells in the same way as intracellular miRNAs. Notably, since exosomes consist of a lipid bilayer, they are stable in various body environments, which makes them a convenient object for being investigated as PCa liquid biopsy biomarker. Additionally, it was demonstrated that PCa cells secrete more exosomes in comparison with normal prostatic cells, which is another argument in favor of investigating techniques for isolating and analyzing exosomes as a perspective approach for PCa liquid biopsy [72].

MiRNAs are short non-coding RNA sequences from 17 to 25 nucleotides in length, regulating gene expression at the post-transcriptional level [73]. Aberrant expression of various specific miRNAs has been observed in various tumors, including breast [74,75], lung [76], colorectal cancer [77], ovarian [78], and prostate cancer [79]. This feature allows not only to distinguish tumor tissue from normal tissue but also to classify tumors according to the tissue origin, and to perform tumor staging [80], which potentially makes miRNAs a biomarker efficient for prognosing the development of oncological diseases [81]. Further, it has been shown that dysregulated miRNAs contribute to tumor initiation and progression by activating a so-called onco-miRNAs or disactivating a so-called tumor-suppressing miRNAs. It was also revealed that one miRNA may have opposite functions, depending on a type of cancer, which indicates the possibility of participation of the same miRNA in different signaling pathways [82,83,84]. Thus, among all types of exosomes, those that are loaded with miRNAs are considered the most promising for being introduced into a clinical practice as biomarker for early diagnosis and prognosis of PCa. Moreover, it was demonstrated that miRNAs presenting within exosomes have higher diagnostic and prognostic value in comparison with the miRNA-free exosomes [85].

Various types of exosomal miRNAs, which may potentially be used as biomarkers in diagnosing PCa at its early form, have been found in blood of patients with PCa. Thus, Bryant et al., compared exosomal miRNAs isolated from plasma from patients with PCa (*n* = 47) and healthy controls (HCs) (*n* = 28), indicating that both miR-107 and miR-574-3p were upregulated in patients with PCa compared with HCs [86]. The role of miR-141 as an onco-miRNA in PCa was further supported by a more recent study involving patients with localized PCa (*n* = 20), patients with BPH (*n* = 20), and HCs (*n* = 20) [87]. The result of this study confirmed the potential value of miR-141 as a diagnostic biomarker for differentiating localized PCa from BPH. Additionally, in the study by Endzeliņš et al. [85], free and exosomal miRNAs isolated from blood plasma were compared in 50 patients with PCa and 22 patients with BPH. As the result, four miRNAs with diagnostic potential were found (miR-375, miR-200c-3p+ miR-21-5p, Let-7a-5p, miR-141). At the same time, in the study by Samsonov et al. [88], no significant differences in miR-141 blood plasma concentration between PCa patients and HCs were identified. According to other results of the study, miR-574-3p, miR-141-5p, and miR-21-5p were identified to be overexpressed in exosomes of 10 PCa patients versus 10 HCs.

It was also proven that the use of a combination of two or more exosomal miRNAs often provides more specific and precise results in PCa diagnostics. According to the study by Foj et al. [89], it was determined that the combination of miR-21 and miR-375 extracted from the urinary exosomes may potentially be more efficient in distinguishing patients with localized PCa from healthy males, compared with the use of each of these miRNAs independently. However, the results were limited by the size of the control group (*n* = 10) and the localized PCa group (*n* = 3). Another example of miRNAs combination, with may potentially be highly valuable in PCa diagnostics, is miR-217 and miR-23b-3p, reported by Zhou et al. [90]. In this study, the blood plasma of patients, including those with early PCa and a Gleason score (GS) at <9, was analyzed for all possible types of the miRNAs. It was demonstrated that 94 different types of miRNAs were expressed at the extent different in PCa patients and HCs. The expression level of 60 of the miRNAs was increased in PCa patients, including the mostly increased miR-217. At the same time, PCa patients had 34 of the miRNAs decreased compared to HCs, including the mostly decreased miR-23b-3p. Further, it was demonstrated by in vitro and in vivo tests, that the increased expression of miR-217 stimulated proliferation and invasion of cancer cells, while increase in the level of miR-23b-3p suppressed it. Nevertheless, a limited group of patients and a short follow-up period necessitates further research in this direction, and verification of the data obtained.

The usage of ratio of one miRNA to another was also proven promising in diagnostics of localized PCa. Thus, Li et al. [91] conducted a study to elucidate the significance of exosomal miR-125a-5p and miR-141-5p, obtained from plasma, as biomarkers for early diagnosis of PCa. The study included 19 HCs and 31 PCa patients. It was demonstrated for patients with PCa that the expression level of miR-141-5p was, on average, only slightly increased, while miR-125a-5p level was significantly decreased. It was also demonstrated, that the miR-125a-5p/miR-141-5p ratio was significantly higher in PCa patients compared to HCs. According to the data presented, the miR-125a-5p/miR-141-5p ratio appeared to perform better as an early PCa biomarker than either of the miRNAs taken individually.

It is worth noting that since recently, the attention for alternative sources of exosomal miRNAs, particularly—semen, has been increasing. In the study by Barcelo et al. [92], the expression level of exosomal miRNAs isolated from semen was analyzed to find an appropriate biomarker for early diagnosis of PCa. The study included PCa patients with moderately elevated PSA levels and GS from 6 to 8 points, patients with BPH and HCs. A total amount of 400 different miRNAs was analyzed within all three experimental groups. According to the results, the most valuable diagnostic potential was found in the combination of PSA serum level, miR-142-3p, miR-142-5p, and miR-223-3p, which demonstrated sensitivity at 91.7% and specificity at 42.9%. Moreover, it was found that miR-342-3p may potentially help to distinguish between samples of patients with GS at 6 and 7 and facilitate to TNM staging of PCa. In the study by Ruiz-Plazas et al. [93], semen samples from patients with PCa at various stages, including the early localized form, were examined to determine the prognostic role of the cytokine tumor necrosis factor-like weak inducer of apoptosis (TWEAK)-regulated exosomal miRNAs. With the miRNA biomarker panel, composed of miR-221-3p, miR-222-3p, and TWEAK, it was possible to classify PCa in terms of its aggressiveness with specificity at 85.7% and sensitivity at 76.9%.

At the same time, despite promising data on diagnostic potential of exosomal miRNAs, it was indicated that diverse exosome isolation techniques may have a significant impact on the results obtained. Thus, in the study by Mercadacal et al. [94], several different techniques of exosome isolation from semen were tested. Their purifying effectiveness and influence on the analysis of miRNAs was evaluated and compared with the results of the standard ultracentrifugation technique. It was demonstrated that techniques originally developed for exosome isolation from blood and urine were also suitable for semen, though the results could vary depending on the method used.

Studies of exosomal miRNAs isolated from various biological fluids have revealed several biomarkers with potential value for PCa prognosis. According to the study by Endzeliņš et al. [85], the level of exosomal miR-let-7a-5p could be helpful for staging PCa in patients with GS of ≥8 versus ≤6. Thus, it was demonstrated by the analysis of exosome samples derived from the patients’ blood plasma (26 with GS ≤ 6 and 24 with GS ≥ 8), that the expression of miR-let-7a-5p was significantly lower in patients with a higher GS (≥ 8) compared to the patients who had lower GS (≤6). In contrast, in the study by Watahiki et al. [95], let-7a demonstrated limited value for indicating the difference between patients with metastatic castration-resistant prostate cancer (mCRPCa) and localized PCa. In addition, in the study by Foj et al. [89], no significant differences in urinary exosome profiles of miR-let-7a-5p were identified between the low-risk and high risk PCa patients. Thus, the role of miR-let-7a-5p as a predictive biomarker of PCa remains controversial and requires further investigation. Additionally, it was demonstrated that the concentration of miR-1246 in urine has an association with the aggressiveness of PCa. According to the results of the study by Bhagirat et al. [96], the miR-1246 expression level could not only be helpful to differentiate between HCs and PCa patients, but also might serve as a predictor of metastases with specificity at 100% and sensitivity at 75%. In another study by Bhagirat et al. [97], exosomal miR-4287 was investigated in terms of its potential to predict the possibility of metastasis at an early stage of PCa. According to the results of this study, specificity of the proposed method at 88.24% was demonstrated. Further, in the study by Guo et al., the potential of six different exosomal miRNAs to predict the development of castration-resistant prostate cancer (CRPCa) was investigated [98]. It was demonstrated that MiR-423-3p was associated with the development of CRPCa. 

In addition to the studies where single exosomal miRNAs were investigated, it was also identified that a combination of several miRNAs may also provide useful prognostic data. Thus, in the study by Huang et al. [99], a combination of exosomal miR-1290 and miR-375 in urine was investigated in terms of the capacity to predict the overall survival rates of patients with localized PCa. It was demonstrated that patients with high concentrations of both miRNAs had overall mortality rates at around 80%, while patients with average or low concentrations of both miRNAs had a mortality rate at 10% over the same follow-up period. The obtained data necessitates further evaluation of the identified miRNAs in a larger cohort of patients. The overall data on diagnostic and prognostic potential of diverse miRNAs is presented in Table 1.

## 5. Circulating Tumor Cells (CTCs)

Originating from a primary or secondary tumor site, CTCs are the tumor cells that have shed into the vessels of the circulatory system or ejaculatory ducts of the prostate gland or urethra [14]. CTCs may be released passively or as a result of active invasion with subsequent intravasation of the CTCs. The CTCs can then be translocated to distant parts of the body to form metastases. During the process of successful dissemination within a human body, CTCs must also overcome the hydrodynamic stress of circulation and evade the host’s immune system, as well as adapt to a new microenvironment to proliferate and colonize [102]. Since the first discovery of CTCs by Thomas Ashworth in 1869 [103], tremendous progress has been achieved in the field of CTCs as a diagnostic and prognostic marker of cancer, including non-metastatic PCa. A significant rise of attention to the field has occurred during the last decade, which is primarily associated with the progress in the development of microelectromechanical systems for isolation and detection of CTCs, their qualitative and quantitate analysis (Figure 4).

Quantitatively, CTCs may potentially provide an opportunity to determine the presence of PCa, its stage, and response to treatment. Initially, most of the studies in the field of liquid biopsy were related to enumeration of CTCs detected in blood samples, which still attracts significant attention. An approach based on detection and enumeration of CTCs has been considered promising in PCa diagnosis and prognostics although clear evidence has not been reached in this context so far. CTCs are considered the precursors of metastases and their detection at early stage of cancer development may be a marker aiding decision making between active surveillance and surgery [104]. In the diagnosis and prognostics of localized PCa, CTCs have been primarily investigated as a parameter for pathological risk stratification. It is anticipated that, as a diagnostic and prognostic parameter, quantifying CTCs may successfully substitute other conventional parameters such as PSA blood level, and GS. Additionally, a correlation between the quantity of CTCs and other clinicopathological parameters is of a great interest.

Qualitatively, isolated CTCs are an object for characterization by means of diverse approaches, including biomarker assay, gene expression analyses, and cultivation with subsequent drug testing and identification of possible drug resistance mechanisms. With biomarker assay and gene expression analysis, it is possible to perform antigen and genetic profiling that allows the identification of molecular subtypes for diverse types of cancer and selection of personalized treatment [14]. At the same time, even though cultivation of CTCs is scientifically attractive, it is technically challenging and is currently far from practical implication in clinic [105].

The techniques of CTC isolation and detection may be in general stratified into two groups: label-free and affinity-based [14]. The affinity-based approaches imply isolation of CTCs by means of adhesion of the CTCs to specific antibodies onto adhesive surface of a liquid-biopsy tool. At the same time, the label-free group includes the techniques that allow to keep the isolated CTCs intact, which may be further subjected to a versatile analysis.

Currently, the potential of label-free techniques for isolating tumor cells from blood of PCa patients has been highlighted in few studies. In the study by Giesing et al., (2010) [106], CTCs were obtained by means of filtration and then detected via RT-PCR analysis for antioxidant genes. As a result, with the described technique, CTCs could have been detected in 42 (32.5%) of 129 patients with localized PCa. Further, in the study by Kolostova et al. [107] where CTCs were isolated from the blood with MetaCell filtration device, 28 (52%) of 55 patients were positive for CTCs and no correlation between the presence of CTCs and GS or T-stage. In addition to the techniques of cell filtration, microfluidics-based techniques of label-free isolation of CTCs have attracted significant attention. Thus, in the study by Todenhofer et al., (2016) [108], microfluidic isolation of CTCs from blood followed by immunostaining with the anti-epithelial cell adhesion molecule (anti-EPCAM) antibody was successful in 25 (50%) of 50 patients, with the amount of isolated CTCs varying from only few to few hundred CTCs among patients. However, no correlation was observed between the CTC detection and clinical parameters of PCa. Considering the results of similar studies, in which CTCs were subjected to the size-based isolation and subsequent immunocytochemical detection, a vast antigen heterogeneity of the CTCs can be inferred [109,110]. Particularly, in the study by Renier et al. [109], it was determined that some of the isolated CTCs did not express keratin (K) while expressing mesenchymal markers such as Vimentin and N-cad, which may signify the involvement of PCa CTCs into the epithelial–mesenchymal transition [111]. In was also demonstrated that liquid biopsy via the size-based microfluidic isolation of CTCs followed by their gene analysis is valuable in terms of prognosis of the disease. Thus, in the study by Miyamoto et al. [112], isolated from blood CTCs were investigated via whole transcriptome amplification, which allowed to predict dissemination of the CTCs to seminal vesicles and lymph nodes In a cohort of patients at 34 subjects. Another approach for label-free isolation of PCa CTCs, gel centrifugation, was proposed by Murray et al. [113]. Notably, in this study, blood was collected from patients at 3 months post-radiotherapy. According to the results of the study, in intermediate-risk PCa participants CTCs were identified at approximately two times more often than in the low-risk participants. Further, it was determined that CTC positive participants had a worse prognosis and shorter time period until biochemical recurrence, and after 15 years of follow-up had a shorter survival rate.

Considering the inconsistent results of diverse studies where blood was used as a source of CTCs, it can be inferred that investigation of other biological fluids in terms of CTC isolation is highly demanded. Thus, in the past few years, CTC isolation from urine has attracted substantial attention. In the study by Nickens et al. [114], urine samples collected from PCa patients were filtered by using Swinney (Sterilitech Corporation, Kent, WA, USA) filtration device, and the cell sediments were further immunocytochemically labeled with anti-ERG, anti-AMACR, and anti-PSA antibodies. According to the results, such approach demonstrated moderate sensitivity and specificity at 64% (16 out of 25 patients) and 68.8% (22 out of 32 HCs), respectively. Further, Campbell et al. [115] investigated urine sediments that were labeled with the MIL-38 monoclonal antibody against the GPC-1 antigen, which has recently been proven specific for several types of cancers, including PCa [116]. The study included 41 patients with localized PCa and 37 patients with benign prostate hyperplasia as a control group. As a result, the technique demonstrated sensitivity and specificity at 71% and 76%, respectively, with the highest specificity at 89% for patients with PSA serum level at ≥4 ng/mL. Later, in the study by Rzhevskiy et al. [117], the same anti-GPC-1 antibody MIL-38 was applied for detecting PCa tumor cells isolated from large volumes (30–100 mL) of urine with a microfluidic chip. Among 14 patients, 12 (86%) contained GPC-1+ putative CTCs in their urine, while urine samples of 11 (79%) out of 14 HCs were free from GPC-1+ cells. The amount of putative tumor cells varied from 4 to 194 cells in patients and was up to 7 cells in HCs. Notably, moderate positive correlation was identified between the amount of GPC-1+ cells per mL of urine and GS. 

As well as in the case of cfDNA, in the case of CTCs semen is the most logical source even though a very limited number of studies have been published in this area so far. For the first time, presence of CTCs in the semen of patients with localized prostate cancer was demonstrated in 1996 by Gardiner et al. [118]. In this study, CTC identification was performed via making seminal fluid smears on the glass slides with further bright-field microscopy. According to the results of the study, 75% of the PCa patients had abnormal cells in their ejaculate while 80% of healthy volunteers were negative for the abnormal cells. However, the technique of making smears accompanied by further bright field microscopy is inconvenient and does not provide reliable cytology. Recently, Rzhevskiy et al. [119] developed the technique of microfluidic label-free isolation of CTCs from the semen. With this technique, it could have been possible to separate approximately 99% of sperm cells and isolate 89% (±3.8%) of tumor cells. In the case of PCa patients, from tens to hundreds of CTCs could have been identified in the seminal fluid while in the case of healthy volunteers there were no or very limited number of abnormal cells in the semen. 

In the majority of studies utilizing affinity-based approaches for CTC isolation, the most well know liquid biopsy platform is the CellSearch System. The technique is based on capturing EpCAM positive cells, followed by CTCs identification as DAPI and Keratin positive, and negative for CD45. However, this concept leads to the enrichment of all circulating epithelial cells including normal cells and may therefore be the cause for the compromised specificity of the technique for detecting PCa CTCs. Namely, in the study by Davis et al. [120], 4 out of 20 (20%) HCs were identified positive for CTCs. However, it is worth noting that an increased amount of blood of 22.5 mL, instead of the standard 7.5 mL, was analyzed in the study. At the same time, in the study by Thalgott et al., (2015) where 20 mL of blood was investigated, none of the HCs was identified positive for CTCs. Furthermore, in other studies in which 7.5 mL blood of healthy controls was investigated, all were CTC negative [121,122]. On the other hand, variable results on the sensitivity of the technique have been reported. Depending on the study, CTCs have been found in blood of 0% to 73% of PCa patients, with varying amounts of CTCs from 0 to 13 per mL of the blood [121,122,123,124,125,126,127,128]. Nevertheless, it is difficult to perform direct comparison between the results obtained in different studies due to the difference in cutoff value of the putative CTCs at which a particular patient or healthy volunteer was considered positive for CTCs. For instance, in the study by Pal et al. [125], the threshold was set at 1 CTC per 22.5 mL of blood, while most studies apply a threshold of 1 CTC/7.5 mL. Finally, in none of the studies using the CellSearch System, a strong correlation between the number of isolated CTCs and PSA serum level, GS score, PFS, or OS was found.

Considering the imperfections of the CellSearch System in terms of CTC detection, mostly due to the EpCAM dependency, alternative affinity-based approaches have been developed in recent years. As two of such approaches, CellCollector and EPISOT were applied in the study by Kuske et al., (2016) [129], and compared with the CellSearch. In the case of CellCollector, CTCs are captured by the EPCAM-based adhesion in vivo to the antibody-coated wire introduced into a peripheral vein of the human body. At the same time, EPISOT assay is the EPCAM-independent technique that is based on the negative depletion of leucocytes with further identification of CTCs as the PSA-positive cells. In this study positivity for CTCs was considered at ≥1 of the tumor cells per 7.5 mL. According to the results of the study, CTCs were identified in blood samples of 58.7%, 54.9% and 37% by the EPISOT, CellCollector and CellSearch assays, respectively. Additionally, significant correlation between the amount of isolated CTCs and PSA serum level or clinical tumor stage, was indicated only in the case of EPISOT, while, in the case of CellSearch and CellCollector, no significant correlations were observed. Another EPCAM-dependent approach based on the microfluidic enrichment of CTCs was tested in the study by Stott et al. [130]. In this study, circulating epithelial cells were identified in blood of 8 out of 17 HCs at the amount of up to 14 cells. At the same time, CTCs were identified in blood of 8 (42%) out of 19 PCa patients, at a relatively high amount of up to 222 CTCs per mL with the median value at 95 CTCs. Furthermore, among the eight patients identified positive for CTCs, six patients had significant decline in the amount of CTCs isolated at 24 h post-prostatectomy. In the study by Russo et al. [131] the immune-magnetic isolation relying on EpCAM and anti-mucin1 antigens was utilized. To identify the enriched CTCs, the immune-magnetic enrichment was followed by the application of several kinds of AdnaTest: (1) AdnaTest ProstateCancer, which detects the overexpression of PSA, PSMA, and EGFR mRNA expression, (2) AdnaTest StemCell, which evaluates expression of transcripts specific for cancer stem cells, and (3) AdnaTest EMT detecting epithelial–mesenchymal transition in the enriched CTCs. Additionally, enriched CTCs were assessed by PCR for AR, c-met, c-kit and thymidylate synthase (TYMS). According to the results of the study, AdnaTest ProstateCancer was positive in 3 (16.7%) out of 18 patients with the low-risk clinically localized PCa and 2 (8.0%) out of 25 patients with the high-risk clinically localized PCa. The AdnaTest StemCell was positive in 5 (27.8%) of 18 and 4 (16.0%) out of 25 patients with the low-risk and high-risk localized PCa, respectively. At the same time, AdnaTest EMT was positive in none of the cases of localized PCa. Moreover, expression of TYMS, AR, c-met, and c-kit by the enriched CTCs was a rare event.

In addition to the EPCAM-dependent alternatives to the CellSearch System, several EPCAM-independent approaches have also been tested. For instance, in the study by Puche-Sanz et al. [132], in which a keratin-based immune-magnetic technique was applied, CTCs at the small amounts ranging from 1 to 4 cells were detected in 16 (18.6%) out of 86 patients. Notably, despite a relatively low amount of CTC positive patients and small numbers of isolated CTCs, a significant correlation was observed between the presence of CTCs in blood and expression of AR in PCa tissue. It is worth noting that in the study by Garcia et al. [133], assessment of AR-V7 splice variant in blood plasma through a capillary nano-immunoassay was proposed as the technique for identifying CTCs. The data from studies where CTCs were investigated as the diagnostic and prognostic marker of non-metastatic PCa are summarized in Table 2.

According to Table 2, the most commonly used affinity-based CTC isolation technique has is the CellSearch System, in which the positivity rate of CTCs in blood of non-metastatic PCa patients ranges between 0–73%, with the number of detected CTCs ranging between 0.0 and 13.3 per mL [114,115,116,117,118,119,120,121,122]. At the same time, in the study where the CellSearch System was compared with such techniques as EPISOT and CellCollector, the techniques demonstrated the least performance in isolating CTCs [123]. Other affinity-based isolation techniques such as AdnaTest [109], multi-Keratin dependent immune-magnetic collection [132], or assessment of AR-V7 through a capillary nano-immunoassay [133], demonstrated a relatively low sensitivity with the highest rate for AdnaTest. Regarding the label-free techniques, the majority did not demonstrate a clinically relevant specificity and sensitivity that would be significantly superior to the standard PCa test. At the same time, in the study where cell filtration was combined with subsequent Keratin and AR dependent immunocytochemistry, the authors reported that CTCs could have been isolated from blood of 100% of the patients [110]. However, it should be considered that blood from healthy donors was not used in this study as a control. Overall, CTC detection from blood for the diagnosis of early-stage PCa has relatively low efficiency. Nevertheless, other biological fluids such as urine and semen are of a high interest for being thoroughly investigated as a potential source of CTCs and alternative to blood.

## 6. Discussion

According to the results of the studies included in this review, all the investigated biomarkers (i.e., ctDNA, exosomal miRNAs, and CTCs) demonstrate potential clinical utility. Several types of miRNAs (Table 1) demonstrated high diagnostic and prognostic value. In addition, it was identified that grouping of two or more miRNAs into the panels could improve sensitivity and specificity in early detection of PCa. Additionally, in several studies, additional conventional parameters, such as the PSA serum level, were added to the miRNA-based biomarker panel, which could increase diagnostic and prognostic accuracy of the panel. Notably, in addition to blood plasma, urine and semen were successfully investigated as the sources of exosomal miRNAs in several studies. In contrast, comparatively moderate sensitivity was demonstrated by ctDNAs as biomarkers for detecting non-metastatic PCa. Even though several promising ctDNA-based biomarker panels were developed, only few of them proof effective. However, ctDNA-based biomarker panels are the least investigated among other types of liquid biopsy biomarkers, and therefore further investigations are required in this field. Regarding CTCs, either weak or moderate sensitivity, specificity, and prognostic value, is demonstrated in the studies where blood was used as the source of CTCs. At the same time, significantly higher diagnostic and prognostic values were identified in rare studies where urine was chosen as the CTC source. 

Although liquid biopsy of PCa by means of isolation and detection of ctDNA, or exosomal miRNAs, or CTCs, is a perspective diagnostic and prognostic technique, it has a number of limitations that have to be considered. The majority of limitations are the inconsistency in protocols, applied for isolation and analysis of the liquid biopsy biomarkers, which leads to inconsistency in the results obtained in different studies. Thus, due to such inconsistency, effectiveness of that or another liquid biopsy approach cannot be reliably compared with alternatives. Therefore, standardization of the liquid biopsy protocols is highly relevant. According to the results of the observed studies, liquid biopsy by means of CTC detection is the mostly standardized compared to other liquid biopsy approaches. Notably, even though the CellSearch System does not detect a high number of CTCs in the blood of patients with non-metastatic PCa, it has already been cleared by the FDA for prognostic use. Additionally, few other platforms for CTC isolation from blood are currently on track for receiving the FDA approval. In the case of miRNAs, as well as ctDNA, the data are limited and do not provide sufficient information to conclude on the diagnostic or prognostic value. One of the most significant issues that has to be thoroughly investigated is the dependance between a timepoint of biological fluid sampling and an outcome of liquid biopsy test. Thus, it was demonstrated that diverse interventions [134], including ultrasound-guided tissue biopsy [135], may cause a release of liquid biopsy biomarkers into the blood stream. Overall, optimization and standardization of methodologies in liquid biopsy of PCa is highly demanded before it could be introduced into a clinical practice. Additionally, while most of the liquid biopsy studies are based on isolation of the biomarkers from blood, such biological fluids as urine and semen are of a great interest for being investigated as the sources of the biomarkers.

Despite the outstanding diagnostic and prognostic value of CTCs in metastatic PCa, several hurdles have to be overcome prior to introducing the technology as a part of medical guidelines for early-stage PCa. For instance, it is known that a common diameter of a single CTC is on average bigger compared to the width of small capillaries [136] and only the smallest and elastic CTCs may release into a systemic circulation, which may be a cause of a relatively low quantity of CTCs in blood at an early stage of tumor development [137]. Additionally, diverse cancers have different typical localizations of metastases. Hence, difference in a quantity of CTCs collected from blood vessels at different sites of a human body is inferred [138], and a conventional minimally invasive blood sampling from cubital vein may not be appropriate in all cases. Further, most of the existing techniques of CTC detection are at their early stage of development and require significant improvement. Particularly, CellSearch^®^ (Menarini Silicon Biosystems Inc, Huntington Valley, PA, USA)—the only FDA-cleared CTC test, is based on EpCAM-dependent immunofluorescent labeling [139]. In this setting, a heterogeneity of CTCs in terms of antigen expression between individuals, and even within a particular individual, should also be considered. Hence, the development of assays composed of multiple markers with an affinity to a particular type of CTCs is needed, which would significantly increase specificity and sensitivity of the corresponding tests. As an alternative to CTC detection via labeling with specific antibodies, profiling of CTCs in terms of specific mutations is considered [140]. Lastly, cultivation of isolated CTCs for tailored medicine is still challenging [141,142]. At the same time, even though more novel techniques of liquid biopsy via isolation and analysis of exosomes or ctDNA are thought to be highly efficient in diagnosis and prognosis of malignant lesions, the cost-effectiveness of liquid biopsy via isolation of CTCs justifies its attractiveness.

## Figures and Tables

**Figure 1 biomedicines-10-03115-f001:**
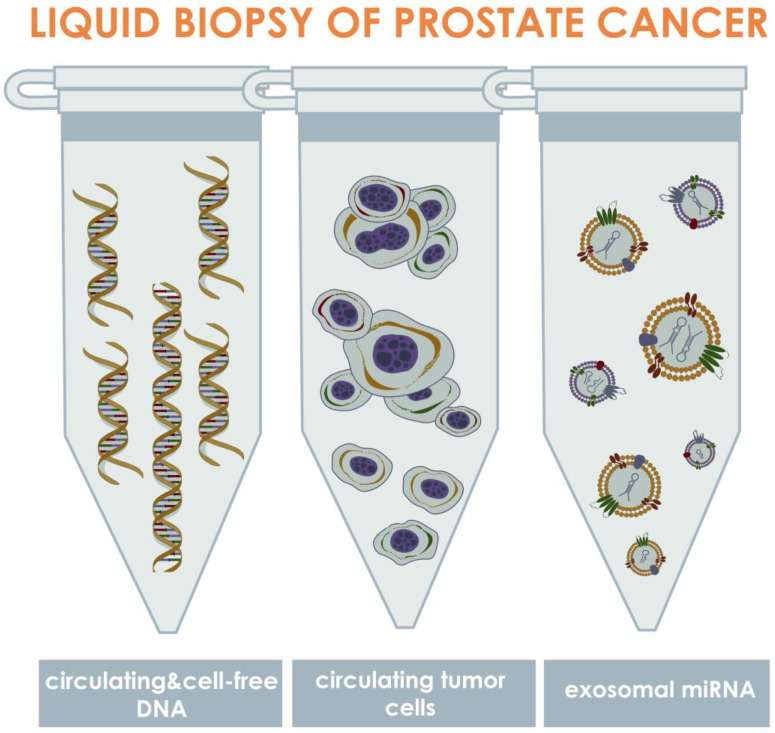
Approaches for liquid biopsy of PCa via isolation and identification of circulating biomarkers. The approaches may be stratified into three primary groups depending on the type of isolated biomarkers, which are: circulating tumor DNA (ctDNA), circulating tumor cells, and exosomal miRNAs. The ctDNAs are the fragments of DNA, released into the bloodstream as a result of tumor cell death or exocytosis; tumor cells are the cells that shed into the biological fluids from primary or secondary tumor sites; exosomes are the microvesicles containing diverse biologically active agents that have been considered biomarkers, including miRNAs presenting in EVs, which release from tumor cells into the biological fluids as a result of cellular metabolism. In PCa, these biomarkers can be mainly found into three biological fluids such as blood, urine, and semen.

**Figure 2 biomedicines-10-03115-f002:**
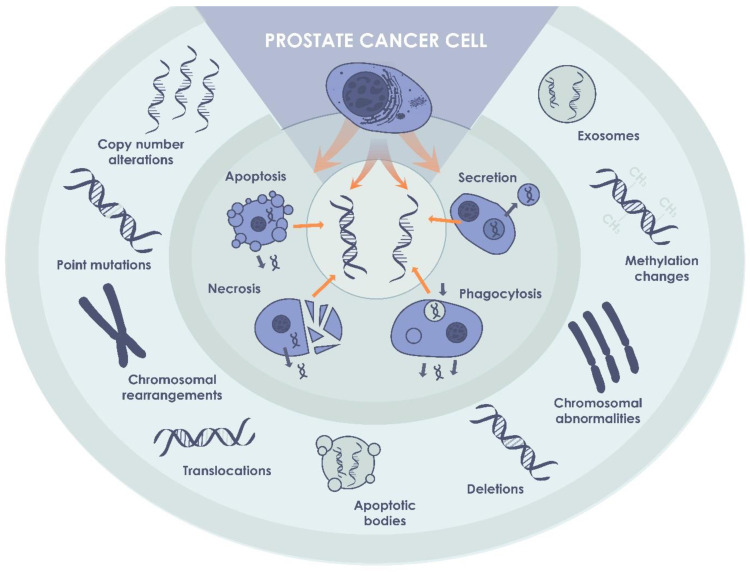
ctDNA in PCa. ctDNA, originating from cancer cells, may be detected in biological fluids of patients with non-metastatic PCa, and can be used as a biomarker of the disease.

**Figure 3 biomedicines-10-03115-f003:**
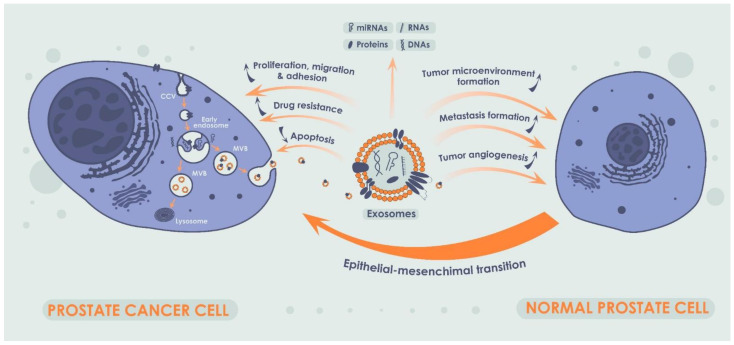
Production of exosomes in cancer cells and their impact on normal prostate cells. A single exosome is formed due to the invagination of the early endosome into the cellular membrane. Exosomes can carry not only miRNAs but also other nucleic acids, a broad spectrum of proteins and lipids, and their combinations thereof. Exosomes, acting as intercellular messengers, are capable of influencing tumor progression by stimulating the proliferation and migration of cancer cells, enhancing cell adhesion, and inhibiting apoptosis. Additionally, exosomes play a crucial role in formation of drug resistance. In addition, exosomes and exosomal miRNA may influence tumor progression by modulating tumor microenvironment, metastasis, and tumor angiogenesis. Notably, exosomal miRNAs in PCa plays a critical role in epithelial–mesenchymal transition.

**Figure 4 biomedicines-10-03115-f004:**
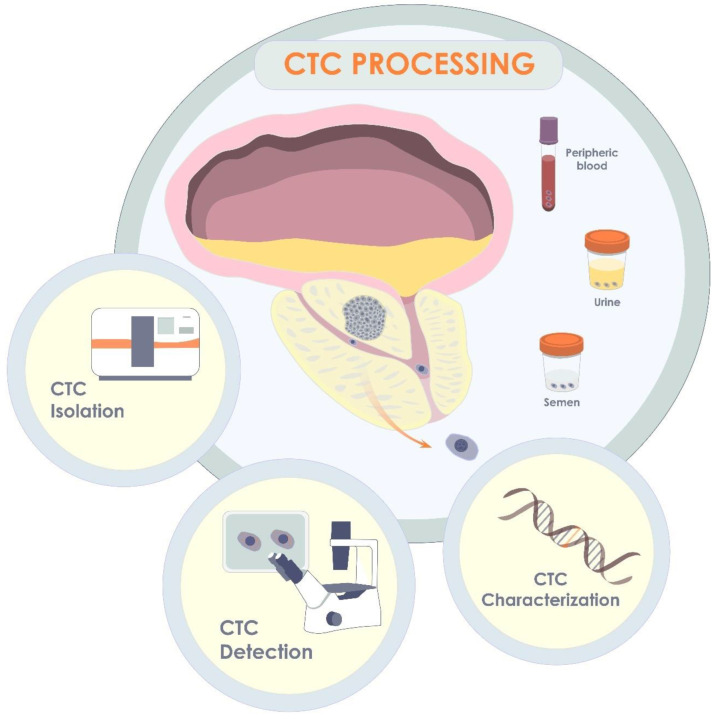
CTCs in localized form of PCa. Originating in primary tumor, CTCs may release into blood, semen, or urine. Thus, being isolated from one of the biological fluids, CTCs may be further detected and quantified, or characterized via molecular analysis, including gene analysis.

**Table 1 biomedicines-10-03115-t001:** Diagnostic and prognostic value of miRNAs in patients with non-metastatic PCa.

miRNA	Biological Fluid	Patients/ Healthy Controls	Isolation Method	Diagnostic Value	Prognostic Value	Reference
miR-342-3p	Seminal fluid	11 HC, 7 BPH, 40 PCa	Centrifugation + microfiltration + ultracentrifugation		Discrimination between GS ≥ 7 and GS = 6 (Sn: 63.6%; Sp: 90%)	[92]
miR-374b-5p					Discrimination between GS ≥ 7 and GS = 6 (Sn: 45.5%; Sp: 90%)	
miR-342-3p+ miR-374b-5p					Discrimination between GS ≥ 7 and GS = 6 (Sn: 54.5%; Sp: 80%)	
miR-142-3p+ miR-142-5p+ miR-223-3p				Discrimination between PCa and BPH (Sn:91.7% Sp:42.9%)		
MiR-375	Plasma	50 PCa, 22 BPH	Size exclusion chromatography	Discrimination between PCa and BPH (N/A)		[85]
miR-200c-3p+ miR-21-5p				Discrimination between PCa and BPH		
Let-7a-5p					Discrimination between GS ≥ 7 and GS = 6	
let-7c	Urine	10 HC, 52 PCa	Differential centrifugation	Discrimination between HC and low-, intermediate-, and high-risk PCa	Discrimination between GS > 8 and GS = 6	[89]
miR-21				Discrimination between HC and low-, intermediate-, and high-risk PCa		
miR-375				Discrimination between HC and low-, intermediate-, and high-risk PCa		
miR-574-3p	Urine	35 HC, 35 PCa	Differential centrifugation + Lectin-Based Agglutination	Discrimination between HC and PCa (Sn: 0.71)		[88]
miR-141-5p				Discrimination between HC and PCa (Sn: 0.66)		
miR-21-5p				Discrimination between HC and PCa (Sn: 0.46)		
miR-196a-5p	Urine	20 PCa, 10 HC	Differential centrifugation	Discrimination between HC and PCa (Sp: 89%, Sn: 100%)		[100]
miR-501-3p				Discrimination between HC and PCa		
miR-217	Plasma	10 PCa, 10 HC	Differential centrifugation + RNeasy Mini Spin kit	Discrimination between HC and PCa		[90]
miR-23b-3p				Discrimination between HC and PCa		
miRNA-125a-5p	Plasma	31 PCa, 20 HC	Differential centrifugation	Discrimination between HC and PCa		[101]
miR-141-5p				Discrimination between HC and PCa		
miR-141	Serum	20 PCa, 20 BPH, 20 HC	ExoQuick Exosome Precipitation Solution	Discrimination between HC + BPH and PCa		[87]
miR-141+ miR-375	Plasma	78 PCa (12 GS<8), 28 HC	Qiagen miRNeasy kit		Discrimination between low-risk and high-risk PCa	[86]
miR-107+ miR-574-3p	Urine		mirVana kit	Discrimination between HC and PCa		
miR-205	Plasma	25 localized PCa, 25 mCRPCa	Differential centrifugation		Discrimination between localized and metastatic PCa, correlation with lower risk of poor outcome	[95]
miR-141					Discrimination between low-risk and high-risk PCa, strong correlation with poor outcome; Discrimination between localized PCa and mCRPCa	
miR-151-3p					Discrimination between localized and mCRPCa	
miR-423-3p					Discrimination between localized and mCRPCa	
miR-152					Discrimination between low-risk and high-risk PCa, strong correlation with poor outcome	
miR-375					Discrimination between localized PCa and mCRPCa	
miR-21					Discrimination between docetaxel-resistant and docetaxel-sensitive PCa patients	
miR-141+ miR151-3p+ miR-16					Discrimination between mCRPC and localized PCa	
miR-126					Discrimination between mCRPC and localized PCa	
miR-1290	Plasma		Differential centrifugation		Discrimination between low-risk and high-risk PCa; association with poor outcome; prediction of androgen deprivation (ADT) failure	[99]
miR-375					Discrimination between low-risk and high-risk PCa; association with poor outcome; prediction of ADT failure	
miR-1290 + miR-375					Discrimination between low-risk and high-risk PCa; association with poor outcome; prediction of ADT failure	
miR-221-3p	Seminal fluid, urine	97 PCa	Differential centrifugation		Discrimination between low-risk and high-risk PCa	[93]
miR-31-5p					Discrimination between low-risk and high-risk PCa	
miR-222-3p					Discrimination between low-risk and high-risk PCa	
miR-193-3p					Discrimination between low-risk and high-risk PCa	
miR-423-5p					Discrimination between low-risk and high-risk PCa	
miR- 221-3p+ miR-222-3p+ TWEAK					Discrimination between low-risk and high-risk aggressive PCa (Sp: 85.7%, Sn: 76.9%)	
miR-1246	Serum	44 PCa	Total exosome isolation reagent	Discrimination between HC and PCa (Sp: 100%, Sn: 75%)	Discrimination between localized and metastatic PCa	[96]
miR-4287	Serum	68 PCa, BPH, HC	Total exosome isolation reagent	Discrimination between HC and PCa (Sp: 88.24%)	Discrimination between localized PCa (GS 4-6) and metastatic PCa (GS > 7)	[97]
miR-423-3p	Plasma	132 PCa, 66 mCRPCa	Differential centrifugation		Discrimination between mCRPC and localized PCa	[98]

**Table 2 biomedicines-10-03115-t002:** Circulating tumor cells in patients with non-metastatic PCa.

Type of CTC Isolation	CTC Isolation Technique (Label-Free or Affinity-Based)	Biological Fluid (Volume)	Number of PCa Patients/HCs and BPH Patients	Cut-Off Number of CTCs for CTC+ Patients	Percentage of CTC+ Patients/Healthy Volunteers	Min-Max Number of CTCs per mL in Patients	Correlation between the Amount or Presence of CTCs and Clinico-Pathological Parameters of PCa	Ref.
Label-free	Filtration + RT PCR for PSA and antioxidant genes	Blood (NA)	129 patients with non-metastatic PCa/NA	NA	32.5%/NA	NA	Strong correlation between the expression of SOD2 or TXNRD1, and tumor size or GS	[106]
	Filtration + CK dependent ICC *	Blood (8 mL)	55 patients with non-metastatic PCa/NA	≥1	52%/NA	NA	No correlation between the presence of CTCs and GS or T stage. No correlation between the CTC count and response to treatment	[107]
	Microfluidics + EPCAM. CK dependent ICC	Blood (2 mL)	50 patients with non-metastatic PCa/NA	≥1	50%/NA	0.5 – 208.5	No correlation between the presence of CTCs and the PSA serum level, age, GS, T stage, or N stage. No correlation between the number of CTCs and GS, T stage, or N stage	[108]
	Filtration + CK, AR dependent ICC	Blood (3 mL)	41 patients with non-metastatic PCa/NA	≥1	100%/NA	NA	NA	[110]
	Microfluidics + droplet digital PCR (dd-PCR)	Blood (20 mL)	34 patients with non-metastatic PCa/34 HCs	NA	NA/NA	NA	d-PCR ** for 8 PCa-specific genes allowed to predict dissemination of CTCs to seminal vesicles and lymph nodes	[112]
	Filtration + ERG, AMACR, PSA dependent ICC	Urine (NA)	25 patients with non-metastatic/32 HCs	≥1	64%/21.2%	NA	NA	[116]
	Sedimentation + GPC-1 dependent ICC	Urine (50–100 mL)	41 patients with non-metastatic PCa/47 HCs and 37 BPH patients	≥ 1	71%/30% and 24%	NA	NA	[115]
	Gel centrifugation + PSA dependent ICC	Blood (8 mL)	241 patients with non-metastatic PCa at 3 months post-radiotherapy/NA	≥1	47.8%/NA	NA	NA	[113]
	Microfluidics + GPC-1 dependent ICC	Urine (30–100 mL)	14 patients with non-metastatic PCa/14 HCs	>8	79%/21%	0.1–4.9	Moderate correlation between the amount of CTCs and GS, or PSA serum level	[117]
Affinity-based	Microfluidics + EPCAM dependent ICC	Blood (20 mL)	19 patients with non-metastatic PCa/17 HCs	≥14 mL^−1^	42%/47%	38–222	Poor correlation between the amount of CTCs and PSA serum level	[130]
	CellSearch	Blood (7.5 mL)	26 patients with non-metastatic PCa/7 HCs	≥1	73%/0%	NA	No correlation between the number of CTCs and GS, or PSA serum level, or T stage	[121]
	CellSearch	Blood (7.5 mL)	10 patients with non-metastatic PCa/NA	≥1	10%/NA	0.0–0.1	NA	[123]
	CellSearch	Blood (7.5 mL)	20 patients with non-metastatic PCa/15 HCs	≥1	5%/0%	0.0–0.1	NA	[122]
	CellSearch	Blood (7.5 mL)	36/NA	≥1	14%/NA	0.0–0.4	NA	[124]
	CellSearch	Blood (22.5 mL)	32 patients with non-metastatic PCa/5 HCs	≥1	45%/0%	0.0–0.1	No correlation between CTC amount and biochemical recurrence of PCa	[125]
	CellSearch	Blood (20.0 mL)	15 patients with non-metastatic PCa/15 HCs	≥1	20%/0%	0.0–0.2	No correlation between CTC detection and GS, PSA serum level, T and N stages, positive surgical margin	[126]
	CellSearch	Blood (7.5 mL)	152 patients with non-metastatic PCa/NA	≥1	11%/NA	0.13–13.3	Weak correlation between CTC detection and GS, PSA serum level, or T stage	[127]
	CellSearch EPISPOT CellCollector	Blood (7.5 mL)	86 patients with non-metastatic PCa/NA	≥1	37%/NA 54.9%/NA 58.7%/NA	0.13–1.3 0.13–1.6 0.13–1.7	Significant correlation between CTC detection and PSA serum values, or clinical tumor stage. No correlations in case of CellSearch and CellCollector	[129]
	CellSearch	Blood (7.5 mL)	59 patients with non-metastatic PCa/NA	≥1	0%/NA	NA	NA	[128]
	Assessment of AR-V7 through a capillary nano-immunoassay	Blood (10 mL)	16 patients with non-metastatic PCa/11 HCs	NA	18.7%/0%	NA	Significant correlation between presence of AR-V7 and expression of CD133 antigen	[131]
	Multi-CK dependent immune-magnetic collection	Blood (10 mL)	86 patients with non-metastatic PCa/17 HCs	≥1	18.6%/0%	0–0.4	NA	[132]
	AdnaTest Prostate-Cancer	Blood (5 mL)	47 patients with non-metastatic PCa/NA	NA	25.5%/NA	NA	NA	[109]

*—immunocytochemistry. **—digital polymerase chain reaction.

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
