# Peer review of "Liquid Biopsy in Diagnosis and Prognosis of Non-Metastatic Prostate Cancer"

_biomedicines, 2022, doi:10.3390/biomedicines10123115_

Round 1

Reviewer 1 Report

Authors of the review described the liquid biopsy in diagnosis and prognosis of non-metastatic prostate cancer. Please find my comments:

-        Picture change the fig 1 to be more informative

-        Please add PRISMA data in the material and methods section

-        Briefly describe the significance of the mentioned genetic alteration (lines 122-124)

-        Add the significance of the method in terms of non metastatic CRPC, including clinical trials

Author Response

Comment: Picture change the fig 1 to be more informative

Response: We have updated the figure. We hope now it looks better.

Comment: Please add PRISMA data in the material and methods section

Response: Our article is more a literature review rather than a systematic review, while PRISMA is referred to the systematic reviews and meta-analyses. The search strategy for this literature review has been described in the “Search strategy” section.

Comment: Briefly describe the significance of the mentioned genetic alteration (lines 122-124)

Response: Dear reviewer, thank you for this advice. Describing each of the six mentioned alterations requires pretty much of text. However, our review is already wide and comprehensive, focusing the aspects of liquid biopsy in non-metastatic PCa. We would like to avoid overloading the review with the evidence on coherent topics. Thus, in a context of the paragraph (lines 123-142), we would better like just to mention the genetic alterations and provide relevant reference, to stay focused.

Comment: Add the significance of the method in terms of non metastatic CRPC, including clinical trials

Response: We have performed an additional thorough literature search and, unfortunately, not much of evidence could has been found in a context of using CTCs, ctDNAs or miRNAs for diagnosis/prognosis of non-metastatic CRPC. The evidence which could has been found, has was included in Table 1.

Reviewer 2 Report

The manuscript by Rzhevskiy provided a contemporary and comprehensive review of the current evidence and understanding of utilization of liquid biospy (including cfDNA, exosome miRNA and CTC) in the diagnosis of localized prostate cancer. It is very well researched, clearly written and will be of interest to a wide readership including GU oncologist and translational/basic researchers. The text is supplemented with tables that provided a nice summary of biomarkers discussed with corresponding references. I applaud the authors' successful efforts in completing this manuscript. 

One very minor comment

In line 110, please change "circulating DNA" to "circulating tumor DNA" to make the description more accurate.

Author Response

Comment: The manuscript by Rzhevskiy provided a contemporary and comprehensive review of the current evidence and understanding of utilization of liquid biospy (including cfDNA, exosome miRNA and CTC) in the diagnosis of localized prostate cancer. It is very well researched, clearly written and will be of interest to a wide readership including GU oncologist and translational/basic researchers. The text is supplemented with tables that provided a nice summary of biomarkers discussed with corresponding references. I applaud the authors' successful efforts in completing this manuscript.

Response: We are grateful for this comment.

Comment: In line 110, please change "circulating DNA" to "circulating tumor DNA" to make the description more accurate.

Response: Has been done.
